

# Modeling Error Learning based Post-Processor Framework for Hydrologic Models Accuracy Improvement

Rui Wu[1], Lei Yang[1], Chao Chen[2], Sajjad Ahmad[3], Sergiu M. Dascalu[1], and Frederick C. Harris, Jr.[1]

[1]Department of Computer Science & Engineering, University of Nevada, Reno
[2]Department of Geosciences, Boise State University
[3]Department of Civil and Environmental Engineering and Construction, University of Nevada, Las Vegas

**Abstract.** This paper studies how to improve the accuracy of hydrologic models using machine learning models as post-processors and presents possibilities to reduce the workload to create an accurate hydrologic model by removing the calibration step. It is often challenging to develop an accurate hydrologic model, due to the time-consuming model calibration procedure and the non-stationarity of hydrologic data. Our findings show that the errors of hydrologic models are correlated with model inputs. Thus motivated, we propose a modeling error learning based post-processor framework by leveraging this correlation to improve the accuracy of a hydrologic model. The key idea is to predict the differences (errors) between the observed values and the hydrologic model predictions by using machine learning techniques. To tackle the non-stationarity issue of hydrologic data, a moving window based machine learning approach is proposed to enhance the machine learning error predictions by identifying the local stationarity of the data using a stationarity measure developed based on Hilbert-Huang transform. Two hydrologic models, the Precipitation-Runoff Modeling System (PRMS) and the Hydrologic Modeling System (HEC-HMS), are used to evaluate the proposed framework. Two case studies are provided to exhibit the improved performance over the original model using multiple statistical metrics.

## 1 Introduction

### 1.1 Motivation

Hydrologic models are commonly used to simulate environmental systems, which help to understand the water systems and their responses to external stresses. They are also widely used in scientific research for physical-process studies and environmental management for decision support and policy-making Environmental Protection Agency (2017). One of the most important criteria for model performance evaluations is prediction accuracy. A reliable model is able to capture the hydrologic features with robust and stable prediction. However, it is challenging to develop a reliable hydrologic model with low biases and variances. In this paper, we aim to develop a post-processor framework to improve the reliability of hydrologic models.

Hydrologic models are typical environmental models for hydrologic process studies and water resources evaluations. Among all types of hydrologic models, physically based parameter-distributed hydrologic models have become increasingly prevalent as they are able to capture detailed features within hydrologic systems. However, in regions with high hydrologic heterogeneities, a large number of parameters are required to represent both temporal and spatial variation. This requests a large





amount of computational resources, which substantially increases difficulties in a model development, data assimilation, and model calibration Ye et al. (2014). The resulting high cost of computation makes it challenging to implement data assimilation techniques such as Ensemble Kalman Filters Slater and Clark (2006); Liu et al. (2016), or use an optimization method such as Shuffled Complex Evolution Duan et al. (1992, 1994). On the other hand, the post-processing methodology dealing with model

results can potentially mitigate such computation requirements and improve the performance Ye et al. (2014). Therefore, the post-processor approach is studied and used in this paper. By studying many hydrologic scenarios, we observe that the hydrologic model errors often follow some patterns that are highly correlated with model inputs (see Fig. 3). Such patterns can be learned via machine learning (see Section 2) and applied in predictions. Thus motivated, we propose a machine learning based post-processor framework that can learn the modeling error to enhance the prediction accuracy.

Despite the potential improvement brought by machine learning techniques, it is worth noting that pure machine learning techniques cannot completely replace hydrologic models. When we compare the performance of the environmental model and machine learning methods, it turns out that the accuracy of the Precipitation-Runoff Modeling System (PRMS) Leavesley et al. (1983); Markstrom et al. (2005, 2015) is much higher than that of commonly used machine learning techniques (e.g. random forest tree Breiman (2001) and gradient-boosted tree Hastie et al. (2009)). Compared to hydrologic models developed using

domain knowledge, pure machine learning models with limited training data cannot accurately characterize all the features of the underlying physical process. Nevertheless, based on hydrologic simulation machine learning approaches are able to further enhance hydrologic model results, by predicting the original modeling errors via learning the relationships between model inputs and output simulation results. In the hydrologic modeling results, the term "simulations" is widely used for both concepts of historical records replication and future prediction.

## 20  1.2   Major Contributions

In this paper, we develop a modeling error learning based post-processor framework to enhance the prediction accuracy of hydrologic models. Based on the results in Section 3, the proposed framework can ease the parameter tuning processes and achieve accurate predictions. The key idea is to leverage the correlation between the hydrologic model inputs and model output errors. There are two main challenges of building the proposed framework: 1) how to improve the efficiency and accuracy in a

hydrologic model in terms of model simulation and development and 2) how to deal with the non-stationary hydrologic data. To solve the first challenge, we propose a machine learning based post-processor, which can capture and characterize model errors to improve hydrologic model predictions. This can help to avoid the misleading effects of irrelevant model inputs. Also, we propose to clean and normalize the data, which enable better characterization of the correlation. To solve the second challenge, we propose a window size selection method, which identifies local stationary regions of the data by using a stationarity measure

based on Hilbert-Huang transform (HHT) Huang et al. (1998). The key idea is to first find all possible window sizes by using data autocorrelation and then select the best window size, which contains the most stationary data. The stationarity measure is proposed to calculate the data stationarity within a window. The two major contributions of this paper are summarized as follows:



- A machine learning based post-processor framework is developed to improve the prediction accuracy and flexibility of hydrologic models. One common issue of existing hydrologic simulation studies is that the development of hydrologic models, in terms of calibration processes, often requires long research time cycles but ends up with barely-satisfied model accuracy. To tackle these challenges, the proposed framework can significantly simplify the parameter tuning processes by learning and calibrating the modeling error using machine learning techniques. Moreover, the proposed framework can use different machine learning methods for different scenarios to obtain the best results, and the model parameters can be dynamically updated using the latest data. Our experiment results in Section 3 show that our method can significantly improve the prediction accuracy, compared with the simulation results of existing hydrologic models.

- A moving window based machine learning approach is proposed, which can enhance the performance of the machine learning technique when dealing with non-stationary hydrologic data. We observe that the distribution of hydrologic data changes over time and the data exhibits seasonality (see Fig. 2). The proposed moving window based machine learning approach can characterize the time-varying relationship between the model inputs and model output errors. The key step is to choose a suitable window size, within which the data is stationary, as most machine learning techniques are designed for stationary data. By leveraging recent advances in the field of nonlinear and non-stationary time series analysis, particularly HHT, we propose the degree of stationarity to measure the local stationarity of the data. Based on the degree of stationarity and the autocorrelation, we propose a window size selection method to optimize the performance of the machine learning techniques.

The proposed framework has been evaluated on the basis of different hydrologic models. The framework can improve the accuracy of the original hydrologic models and the window selection method can find the data pattern and select a suitable window size. Moreover, we find that the accuracy of an uncalibrated hydrologic model is as good as the calibrated one by using the proposed framework, which indicates that the proposed framework can replace the complicated "calibration" step in the traditional hydrologic model developing workflow. Section 3 introduces more details of the case studies.

## 1.3 Related Work

An appropriate window size is very important for training a machine learning model to deal with non-stationary time series data. Most of the existing work on the window size selection is based on concept drifts and distribution changes. There are some methods that perform well but can only be applied to a certain machine learning method, such as Klinkenberg and Joachims (2000); Bifet Figuerol and Gavaldà Mestre (2009). Bifet Figuerol and Gavaldà Mestre in Bifet Figuerol and Gavaldà Mestre (2009) proposed a concept drift based method to dynamically adapt window size for Hoeffding Tree Domingos and Hulten (2000). To solve the limitation, some methods are proposed that can be applied to different machine learning techniques by using statistical techniques to monitor the concept drifts. In Klinkenberg and Renz (1998); Lanquillon (2001); Bouchachia (2011), Statistical Process Control (SPC) Oakland (2007) is leveraged to monitor the data change rate by using error rate. If the error rate change is larger than a threshold, it means the data is not stable, and then the window size should be changed. These methods need to assume that the error rate follows a certain distribution, and then calculate the threshold by using the



error confidence interval. Similarly, in Gama et al. (2004); Bifet and Gavalda (2007), window selection methods are proposed based on the concept of context with the stationary distribution. The proposed methods require the dataset inside a window to follow a certain distribution, and then calculate the confidence interval by using an approximate measurement Gama et al. (2004); Bifet and Gavalda (2007). However, this requirement may not be satisfied for some hydrologic data because the data distributions may be not known or follow a certain distribution. Different from these works, we choose the window size based on the degree of stationarity of the data, based on the proposed stationarity measure (see Section 2.2.2), which does not assume the data follows any predetermined distribution and is applicable to different machine learning techniques.

There are many methods to improve the performance of hydrologic model simulations by reducing uncertainties from various sources: model input pre-processing, data assimilation, model calibration, and model result post-processing Ye et al. (2014). Model input pre-processing deals with uncertainties from model input variables such as establishing precipitation measurement networks or post-processing meteorological predictions Glahn et al. (2009). Data assimilation treats the uncertainties from model initial and boundary conditions. For instance, the assimilation of snow water equivalence data can improve initial conditions in a snow or hydrologic model Andreadis and Lettenmaier (2006); Slater and Clark (2006). Model calibration technique reduces the uncertainty from model parameterization Duan et al. (1992, 2006), such as using a transformation of model residuals to improve the model parameter estimations Safari and De Smedt (2015), or using optimization algorithms to find best parameters that fit the observations Hay and Umemoto (2007); Skahill et al. (2009). Post-processing quantifies and reduces the uncertainties related to model results. Statistical models are usually used for post-processing, which calculates the conditional probability of the observed flow given forecast flow Ye et al. (2014); Seo et al. (2006). Examples include variants of Bayesian frameworks built on model output Krzysztofowicz and Maranzano (2004), the meta-Gaussian approach Montanari and Brath (2004), the quantile regression approach Seo et al. (2006), and the wavelet transformation approach Srivastava et al. (2009). Because the post-processing methodology only deals with model results it requires less computations for most cases. Therefore, we propose to use the post-processing method in this framework.

There are many different post-processing approaches being used for hydrologic modeling. According to Brown and Seo (2013), the existing algorithms generally varies, in terms of: 1) the source of bias and uncertainties; 2) the way of predictor developed using prior available data; 3) the assumptive relationship between predictors and model simulations; 4) the uncertainty propagation techniques; 5) the model method used in spatial, temporal, and cross-dependencies simulation; and 6) the parameterization means. Specifically, Zhao et al. (2011) introduced a general linear model, which leveraged and removed the mean bias from the original model outputs, to improve the original model predictions. Quantile Mapping (MQ) method was used as an effective method, which uses Cumulative Density Functions (CDFs) of observations and simulations to remove corresponding differences on quantile basis AW and DP. (2006); T et al. (2006). Based on this, Madadgar et al. (2014) proposed a couple equations of univariate marginal distributions joint CDFs that further improved the representation of the inherent correlations between observations and simulations, and the separation of the marginal distribution of random variables. Brown and Seo (2010) designed an advanced data transformation method for nonparametric data using Conditional Cumulative Density Function (CCDF) Schweppe (1973), which has been successfully applied to nine eastern American river basins Brown and Seo (2010). Krzysztofowicz and Maranzano (2000) proposed a Bayesian based methodology using normal quantile transform in a





Meta-Gaussian distribution as a way to remove model biases. However, these methods that rely on the original model calibration are limited to the applied basins Zhao et al. (2011), variable uncertainties, the static dataset in use, and instabilities by data outliers and "ancient" dataset Brown and Seo (2013). which can substantially reduce the result performance and reliability of the post-processing algorithms.

The rest of this paper is organized as follows. In Section 2, the modeling error learning based post-processor is proposed. In Section 3, two case studies are presented and the results are analyzed. In Section 4, the discussions of our study are provided. The paper is concluded in Section 5.

## 2  Modeling Error Learning Based Post-processor Framework

Hydrologic models are based on the simulation of water balance among principal hydrologic components. With different study
purposes, the selected hydrologic model varies and so as the parameters used in the simulation algorithm. It is challenging to develop an accurate hydrologic model and traditional hydrologic models can often have high biases and variances in the outputs. By studying many hydrologic scenarios, we observed that the hydrologic model errors often follow some patterns that highly correlate with the model inputs and such patterns can be learned via machine learning. Thus motivated, we propose a machine learning based post-processor framework that can learn the modeling error to enhance the prediction accuracy. The
details of the proposed framework are provided in the follow section.

### 2.1  Observations and Motivations

We study the prediction errors of a PRMS model Leavesley et al. (1983); Markstrom et al. (2005, 2015) using 10-year historical watershed data collected from USGS USGS (2017). The study area is the Lehman Creek watershed in eastern Nevada and the data is collected every 24 hours. Fig. 1 illustrates the error distribution of streamflow prediction from the PRMS model. The
distribution is very close to a normal distribution with a close-to-zero mean value and a low variance. However, when taking a closer look at the prediction errors across time (see Fig. 2), we observe a large discrepancy between the model outputs and the ground truths in the middle of each year. It implies that the current PRMS model cannot accurately characterize the streamflow in the middle of a year. Therefore, there is a need to better capture the dynamics of the streamflow in this time period.

Intuitively, the prediction errors contain important information, which can be leveraged to reduce the hydrologic model
errors, so as to improve the prediction accuracy. Therefore, we explore the information contained in the prediction errors and find that the prediction errors are actually highly correlated with the model inputs. As shown in Fig. 3, during May, June, July, and August of the year 2011, the streamflow prediction errors are highly correlated with the temperatures and time (month and day). The larger correlation values and stars in Fig. 3 in the upper side mean the closer relations between two variables. By leveraging the correlations, we aim to predict the original model errors and thereby improve the prediction accuracy.

Along this line, we propose to use machine learning techniques to learn the modeling errors by leveraging the strong correlations between the prediction errors and the model inputs, in order to improve the accuracy in streamflow predictions. The proposed framework is illustrated in Fig. 4. It mainly consists of three steps:





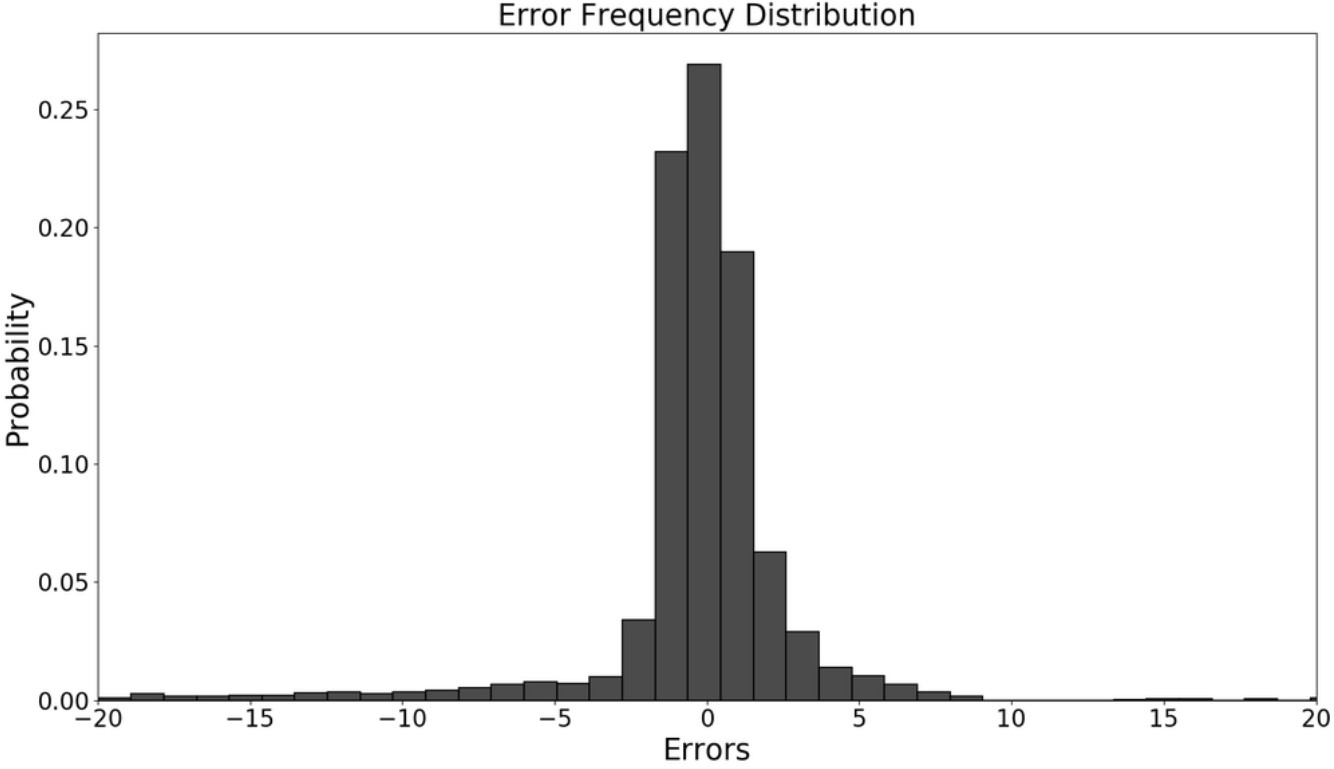

**Figure 1.** A Traditional Calibrated PRMS Model Streamflow Prediction Errors Histogram (Example of the Lehman Creek).

- – Step 1: Develop a hydrologic model, such as PRMS. The model can generate predictions (e.g., streamflow prediction) based on the inputs (e.g., temperature, time, and precipitation).

- – Step 2: Obtain the hydrologic model errors. By comparing the ground truths with the hydrologic model predictions, the framework can collect historical hydrologic model errors.

5 - – Step 3: Preprocess history errors and build a machine learning model. The hidden correlations between the model errors and the model inputs can be enhanced after preprocessing and be characterized by a machine learning model.

After these three steps, the trained machine learning model is integrated with the original hydrologic model to enhance the prediction accuracy. It produces the improved results by adding the predicted errors with hydrologic model predictions. Different methods in each "Preprocessor", "Machine Learning Model", and "Hydrologic Model Errors" component can be 10 selected based on the needs of applications. The details of each component as shown in Fig. 4 are described in the following sections.

**Remarks:** In practice, the development of a hydrologic model needs to be calibrated based on hydrogeologic conditions and meteo-hydrologic characteristics. The calibration procedure is a process that finalizes parameters used in the model numerical



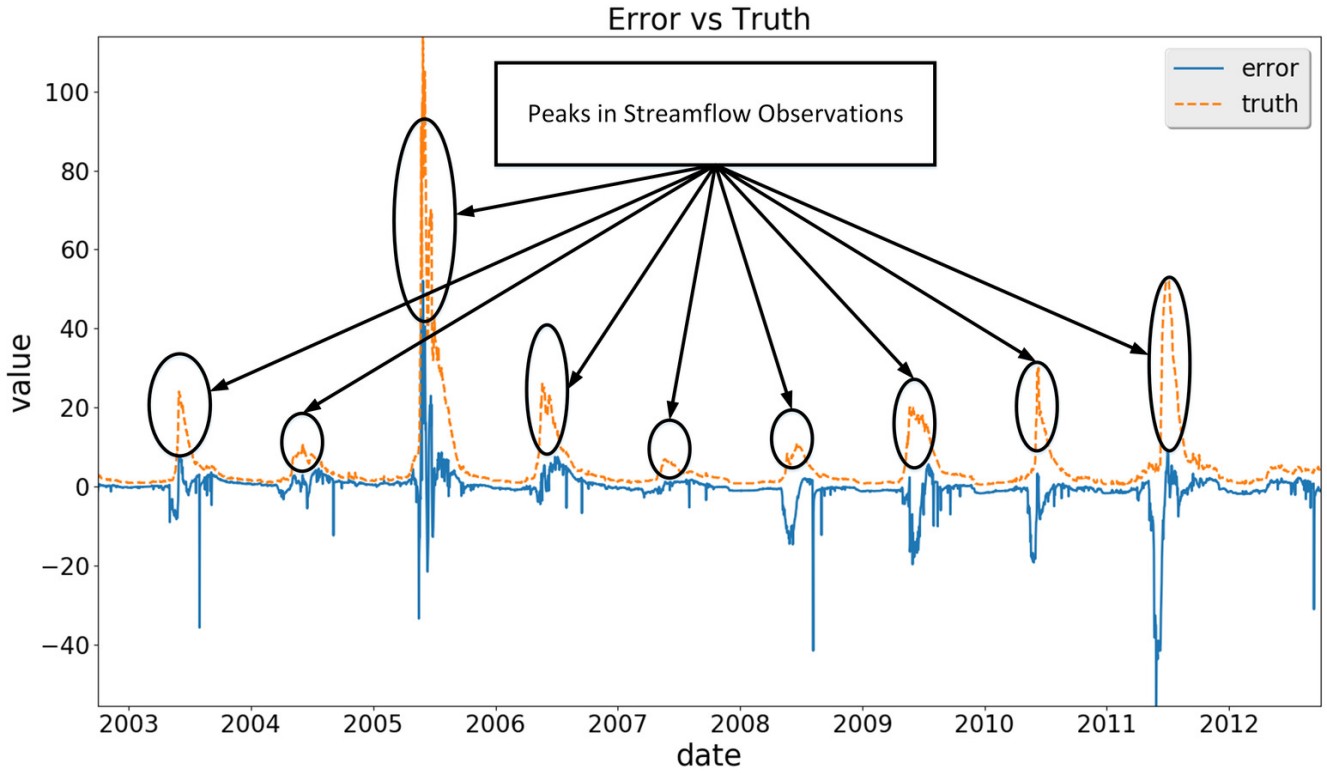

**Figure 2.** Comparisons Between Streamflow Observations and Prediction Errors from a Traditional Calibrated PRMS Model (Example of the Lehman Creek). The y-axis value unite is "Cubic Feet per Second".

equations that determining the hydrologic process simulation. With temporal and spatial heterogeneity, these parameters could either be characterized with both these features, such as in a physically based parameter-distributed hydrologic model PRMS, or be averaged to represent a mean level while still maintaining the capability of capturing the streamflow variation, such as in the Hydrologic Modeling System (HEC-HMS). In this study, the default values of each parameters are used in the un-

5  calibrated cases as to compare with the calibrated cases from traditional hydrologic calibration and post-processor methods. As demonstrated in Section 3, the proposed framework provides a better prediction accuracy when compared with the traditional hydrologic calibration method.

## 2.2 Modeling Error Learning enhanced hydrologic Model

The detailed workflow of the designed modeling error learning enhanced hydrologic model is illustrated in Fig. 5. The basic

10  idea is to use predicted error to calibrate original hydrologic model's predictions as shown in Eq. (1)

$$\hat{p}_t = f(x_t) + g(x_t) \tag{1}$$




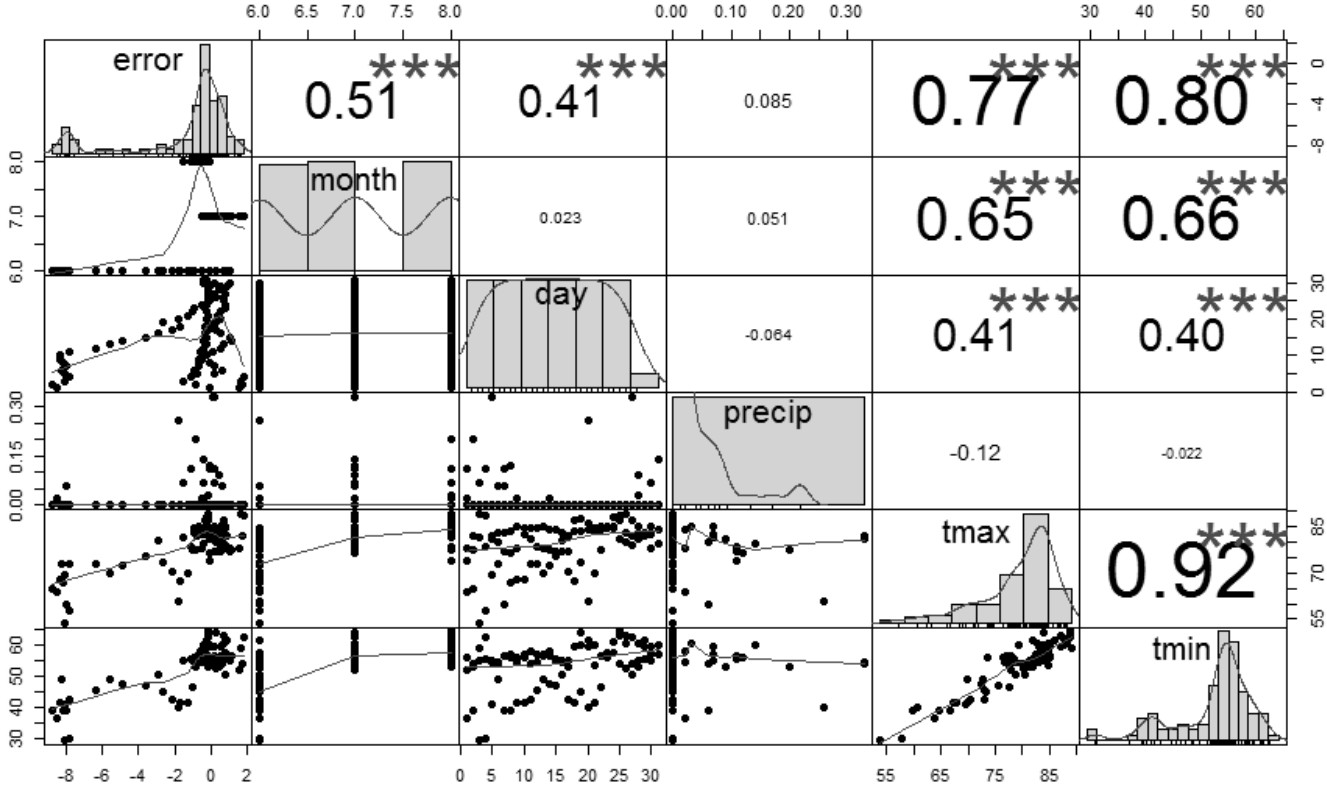

**Figure 3.** Correlations Between PRMS Inputs (i.e. *precip*, *tmax*, and *tmin*) and Streamflow Prediction *Errors*, during May, June, July, and August (2011): The diagonal graphs show the variable distributions, the lower side graphs show the scatter plots between the corresponding row and column variables, and the upper side values are the correlation values between the corresponding row and column variables. (*precip*: precipitation; *tmax*: maximum temperature; *tmin*: minimum temperature); *errors*: streamflow prediction errors.

where $\hat{p}_t$ denotes the improved prediction at time $t$; $x_t$ denotes the model inputs (i.e., temperature, time and precipitation) at time $t$; $f(\cdot)$ denotes the hydrologic model, which generates predictions based on $x_t$; and $g(\cdot)$ denotes the error prediction model learned in the "Machine Learning Model" component, which generates hydrologic model prediction error based on $x_t$.

As illustrated in Fig. 5, there are basically three steps to build an enhanced hydrologic model:

5     – Step 1: Calculate the hydrologic model errors. We calculate errors using differences between the observations and model predictions in the "Hydrologic Model Errors" component.

    – Step 2: Enhance the correlation between hydrologic model errors and inputs. This step contains two sub-steps: "scale model error" and "data transformation". "Scale model error" is used to scale error into a certain scope (e.g. between 0 and 1) and "data transformation" is used to normalize hydrologic model errors and stabilize the variances of hydrologic

10        model errors.





**Figure 4.** The Diagram of Modeling Error Learning Based Model Post-processor Framework.





**Figure 5.** Modeling Error Learning Enhanced Hydrologic Model.

- – Step 3: Build a machine learning model. The scaled and transformed original hydrologic model errors and model inputs are used to train a machine learning model to predict the hydrologic model errors. The predicted errors need to be back-transformed and back-scaled before being used to compensate the hydrologic model results.

More details of the framework components (rectangles in Fig. 5) and steps (arrows in Fig. 5) are introduced in the follow

5  sections.





### 2.2.1 Preprocessor Component

The "preprocessor" component preprocesses the hydrologic model errors, and the outputs of this component are used to train a machine learning model in the "Machine Learning Model" component. The objective of the "preprocessor" component is to normalize errors and reduce error variances. In other words, this component is used to make it easier for the "Machine Learning

Model" component to characterize correlation between the hydrologic model inputs and errors. Specifically, this component scales and transforms the hydrologic model prediction errors using Eq. (2)

$$e_t = tr(\alpha e) \tag{2}$$

where $e_t$ denotes preprocessed error; $tr(\cdot)$ denotes transformation function, $\alpha$ denotes the scaling factor; $e$ denotes the original hydrologic error. Based on the case studies in Section 3, a good scaling factor is often between zero and one.

Note that in this framework different functions can be selected based on the dataset characteristics. For example, if the dataset is positively skewed, log-sinh transformation Wang et al. (2012) could be helpful. If the dataset has a large variance, boxcox transformation Wang et al. (1964) may be applied. In Section 3, Case Study 1 uses the log-sinh transformation (see Eq. 13) and Case Study 2 uses the boxcox transformation (see Eq. 15). These transformation functions can improve the hydrologic model outputs, as shown in Section 3.

**Remarks:** The "Preprocessor" component should be repeated multiple times to find out the best-performed scaling factor and data transformation parameters. For example, %-time cross validation can be used to test all possible parameter combinations' performance Kohavi et al. (1995). The performance can be measured by using RMSE (Eq. 8), PBIAS (Eq. 9), NSE (Eq. 10), or CD (Eq. 11). After a good parameter combination is chosen, it will be used in both the "Preprocessor" component and the "Back-transform and Back-scale" step.

### 2.2.2 Machine Learning Model Component

The "Machine learning model" component aims to predict the transformed hydrologic model error $\hat{g}(x_t)$, using the hydrologic model input $x_t$. To obtain the original model prediction error $g(x_t)$, $\hat{g}(x_t)$ needs to be transformed back using the inverse of the transformation function, which is discussed in Section 2.2.4. In what follows, we discuss how to find $\hat{g}(\cdot)$ using machine learning techniques.

There are many machine learning techniques that can be applied in this component, such as Support Vector Regression (SVR) Basak et al. (2007) and gradient boosted tree Hastie et al. (2009). Most of them are designed for stationary environments, in the sense that the underlying process follows some stationary probability distribution. However, hydrologic processes are often non-stationary. As illustrated in Fig. 2, the streamflow shows seasonality in the sense that the patterns of streamflow in each year are similar but change over time. To address this challenge, we propose to use a moving time window to adapt to the

changes due to hydrologic data variations.

     The basic idea is to set up a time window and train the machine learning model using the data within the window, which moves over time. By using the time window, we are able to track the changing dynamics of hydrologic data. However, it is





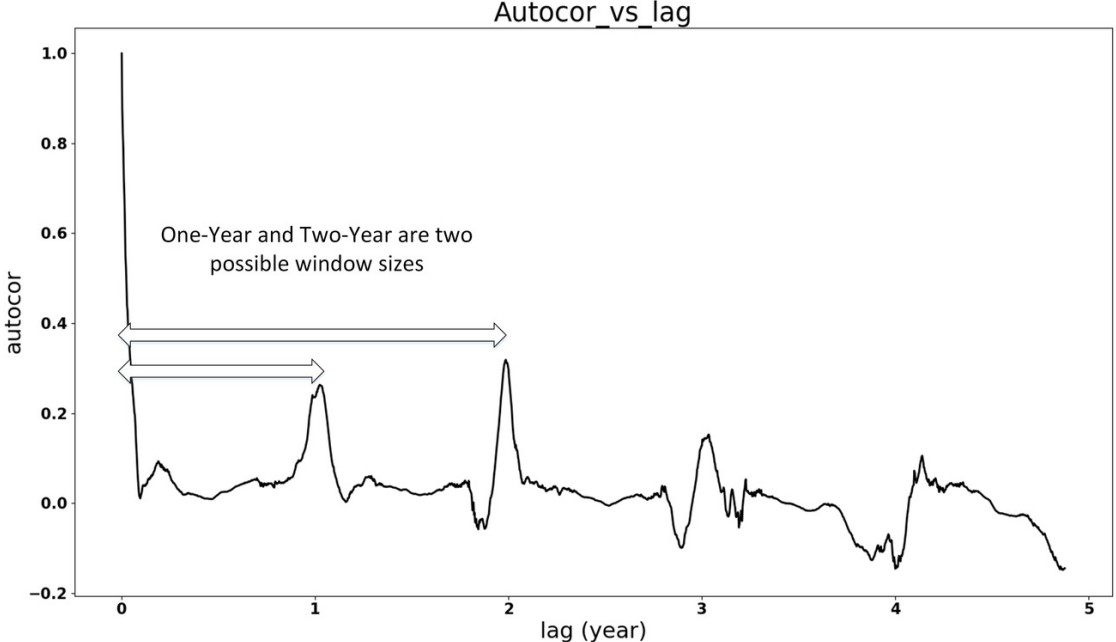

**Figure 6.** Case Study 1 Training Data Autocorrelation Values vs Lag Days: One-year and two-year can be the data pattern lengths. Because these are the distances between the start point and peaks in the training data.

challenging to find an appropriate window size. If the window size is too large, it increases model training complexity and the model is not able to quickly adapt to the changes of the hydrologic data. Even though a model with a large window size may generate accurate results during the training phase, it is possible that the accuracy of the model using the test dataset could be very poor, which is due to overfitting issue Domingos (2012). If the window size is too small, the model may not be able to

5   capture the pattern of the hydrologic model errors.

In this paper, the window size selection is based on the pattern and the degree of stationarity of the data, which can not only capture the data pattern, but also ensure the data stationarity within the window.

To find the data pattern, we leverage the autocorrelation of the data. Due to the seasonality, the autocorrelation shows a peak every year (see Fig. 7) and the distance between two peaks indicates that the pattern repeats during this period. However, as

10   illustrated in Fig. 7, there are several peaks, and it remains challenging to determine the window size, i.e., "how many peaks should be chosen?"

To address this challenge, we further calculate the degree of stationarity of the data in a given window size and use this to determine the window size. Specifically, the degree of stationarity ($DS$) is defined by leveraging recent advances in the field





of nonlinear and non-stationary time series analysis, particularly Hilbert-Huang transform (HHT) Huang et al. (1998). $DS$ is defined as:

$$DS(T) = \frac{\sum_\omega \hat{DS}(\omega)n(\omega)}{n_{sum}} \tag{3}$$

$$\hat{DS}(\omega) = \frac{1}{T}\sum_{t=0}^{T}(1 - \frac{H(\omega,t)}{n(\omega)})^2 dt \tag{4}$$

$$n(\omega) = \frac{1}{T}\sum_{t=0}^{T}H(\omega,t) \tag{5}$$

where $DS(T)$ denotes the data stationarity value of window size $T$ (Eq. 3), $\hat{DS}$ can characterize the variation of the data in a certain frequency ($\omega$) bin over time (Eq. 4), $n(\omega)$ is the average amplitude of the frequency (Eq. 5).

In Eq. 3, $n_{sum} = \sum_\omega n(\omega)$. $DS(T)$ sums $\hat{DS}$ value of each frequency and weights each of them by using $n(\omega)$. This ensures that small, relatively insignificant oscillations do not dominate the metric. $n_{sum}$ in the denominator part normalizes $DS(T)$

and allow different $DS$s to be comparable. Note that the larger $DS$, the more non-stationary the data, and we prefer a small DS in a given time window.

In Eq. 4, $H(\omega,t)$ denotes the Hilbert spectrum, which is a frequency-time distribution of the amplitude of the data. A large $\hat{DS}$ indicates large variations in the bin, which means non-stationary behavior. A close-to-zero $\hat{DS}$ indicates small variations in the bin, which means stationary behavior.

The $\hat{DS}$ concept is first introduced in paper Huang et al. (1998) but it only considers the data stationarity of a certain frequency bin and does not characterize the entire time series data stationarity. To improve the $\hat{DS}$ concept, we propose $DS$ that calculates the whole dataset stationarity.

After the possible data patterns are chosen based on autocorrelation, the data pattern that has the minimum $DS$ (the most stable) is chosen to be the final window size.

Fig. 7 illustrates the values of $DS$ under different window sizes for Case Study 1 in Section 3.2. The $DS$ value increases as the window size grows, which means the data becomes more non-stationary when the window size grows. As the one-year $DS$ is smaller than two-year $DS$, the one-year window size is chosen for the Case Study 1, because it is one of the data patterns and this window size has the minimum $DS$ value. Fig. 8 compares the prediction performance using different window sizes for Case Study 1. It shows the one-year window size has the best performance. In contrast, the 4.5-year window size is more

accurate than one-year window size with the training dataset but the performance is worse with the testing dataset, which means a larger window size can cause overfitting issues.



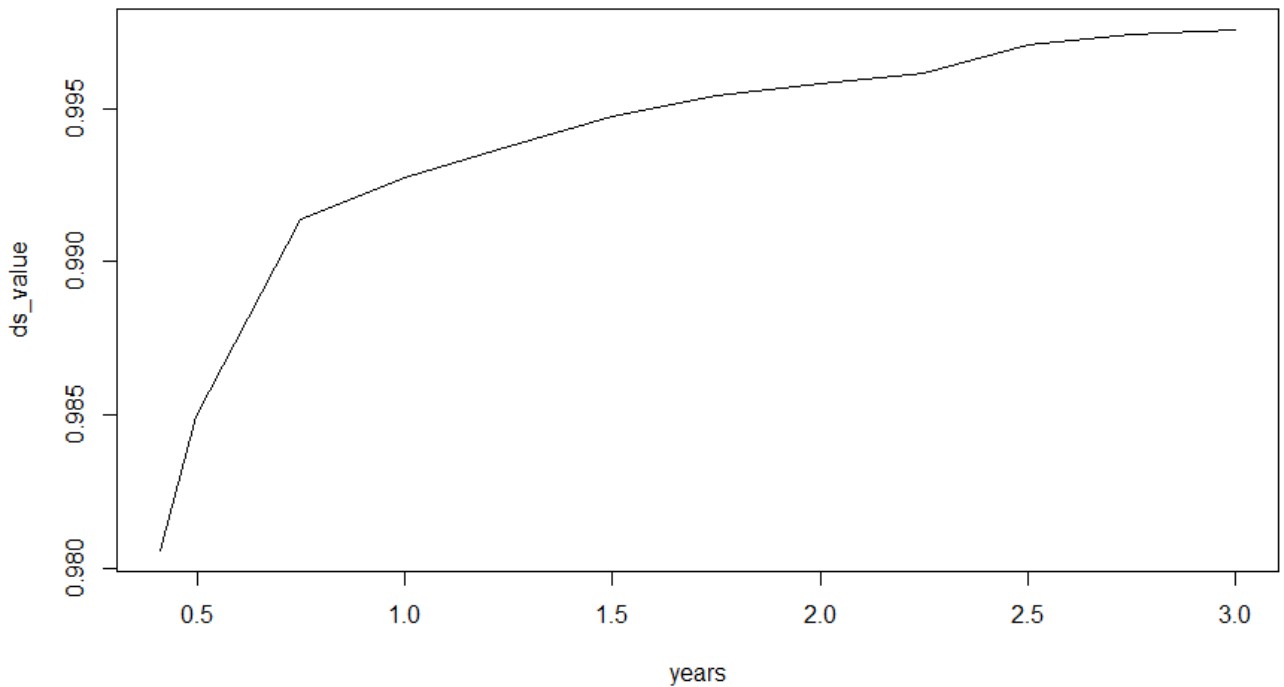

**Figure 7.** Case Study 1 Training Data $DS$ vs Window Size: One-year $DS$ is less than two-year $DS$. This means the one-year window contains more stable data and should be chosen.

### 2.2.3 Back Transform and Back Scale

The predicted errors generated from the "Machine Learning Model" cannot be used directly because the machine learning model is trained with the preprocessed errors. The predicted errors need to be back preprocessed using the corresponding preprocessor methods to obtain the real predicted hydrologic model errors.

5    Let $tr^{-1}$ denote the inverse of the transformation function. $g(x_t)$ can be computed as follows:

$$g(x_t) = tr^{-1}(\hat{g}(x_t))/\alpha \tag{6}$$

And the prediction $\hat{p}_t$ can be given as:

$$\hat{p}_t = f(x_t) + tr^{-1}(\hat{g}(x_t))/\alpha \tag{7}$$





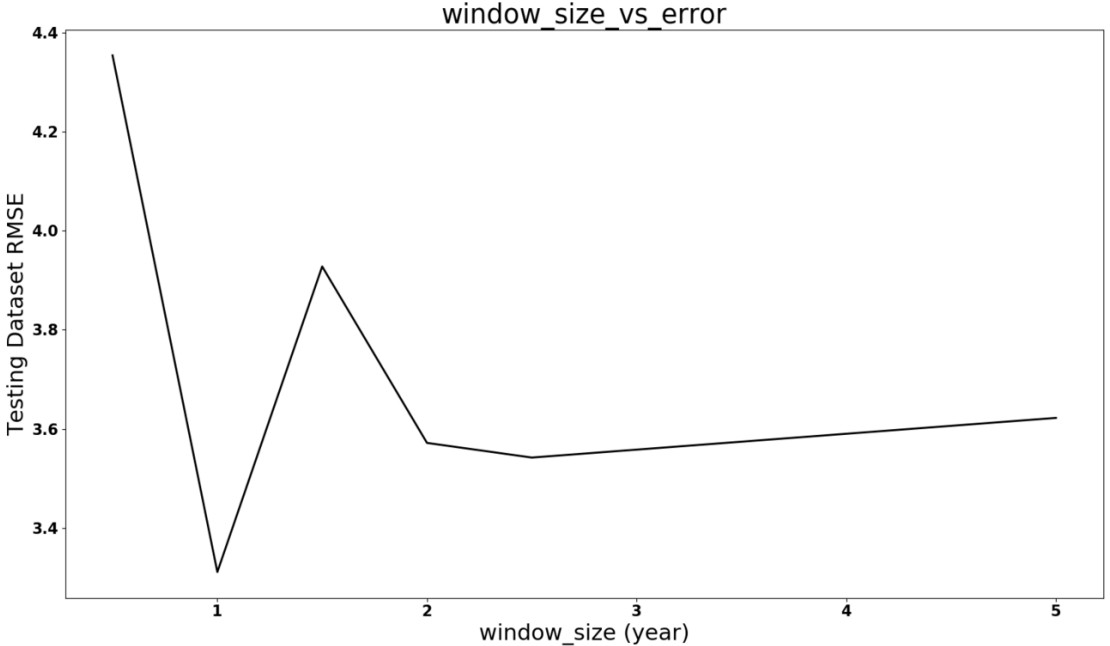

**Figure 8.** Case Study 1 Testing Data RMSE vs Window Size: The one-year window size is better than other window size based on RMSE value.

### 2.3 Discussion of Proposed Methods

The "Modeling Error Learning" is the key component of the framework. If it is able to predict the hydrologic model errors, the framework works. If not, then the framework cannot improve a hydrologic model performance. Therefore, the question "when the framework does not work" equals "when the 'Modeling Error Learning' component cannot predict errors accurately".

5 Because this component leverages the relations between the model inputs and model errors, the component can work when the model inputs are correlated to the model errors. Therefore, a modeler can calculate the correlation values between each model inputs and the preprocessed model errors of the historical data to test if the proposed framework can work. If some model inputs are correlated with the preprocessed model errors, then the proposed framework is able to improve the hydrologic model accuracy and vice versa.

10 "How the framework can perform better" is another important question. It depends on the chosen machine learning techniques used in the "Modeling Error Learning" component. The errors contain biases and variances. Based on bias-variance tradeoff theory Friedman (1997), when bias decreases, variances will increase and vice-versa. Different machine learning techniques have different characteristics. For example, a boosted tree has a high bias, low variance, and performs well when dimensionality is low; A random forest has a low bias, high variance, and performs well when dimensionality is high Caruana



and Niculescu-Mizil (2005). Thus, the selection of machine learning method should be determined by study needs and data characteristics.

However, it is hard to determine which machine learning technique works better for a certain problem before performing tests. We suggest to do a pre-test to exam which machine learning technique could work and perform better. The pre-test data
should be part of historical data and the size is decided by the data cycle, such as a week, month, and year. For example, the temperature is high in summer and low in winter. Therefore, a "year" can be a cycle. The first two years temperatures of the historical data are chosen to be the pre-test data. The first year temperature values are used in the training phase, and the second year temperature values are used in the testing phase.

Hydrologic data can vary dramatically in a short time period, which is hard to be captured by a hydrologic model. It is
also difficult for the "Machine Learning Model" component to accurately predict the hydrologic model errors. To address this issue, we propose a smooth prediction method to regulate the hydrologic model errors are less irregular and therefore enhance the performance of the "Machine Learning Model" component. Fig. 2 is an example of dramatically-changed streamflow. The streamflow observations grow rapidly in the middle of each year and the vibrations generate small spikes along the uphills and downhills. The original PRMS model cannot characterize the spikes and generate irregular errors. Because the "Machine
Learning Model" component is built based on these errors, the framework cannot perform very well in the middle of every year and generate unnecessary peaks. We propose a method to smooth the hydrologic model predictions to avoid the spikes, which contains three steps:

1.  Choose a threshold T, which should be between the maximum and minimum value.

2.  Smooth the hydrologic model predictions by using T. If the difference between the previous prediction and current
20       prediction is higher than T, then we use the previous prediction to replace the current prediction.

3.  Check if the current T avoid peaks. If the current T cannot avoid any peaks, then choose a smaller T, and then go to Step 1. If there is a "plateau" (flat peak) as Fig. 9 displays, then choose a larger T.

When a fitting T is finalized, it is used both in the training phase and in the test phase for the hydrologic model predictions. In the training phase, it can help to choose more appropriate scale factors, transformation parameters, and window size. In the
test phase, it can to avoid original hydrologic model severe vibration predictions.





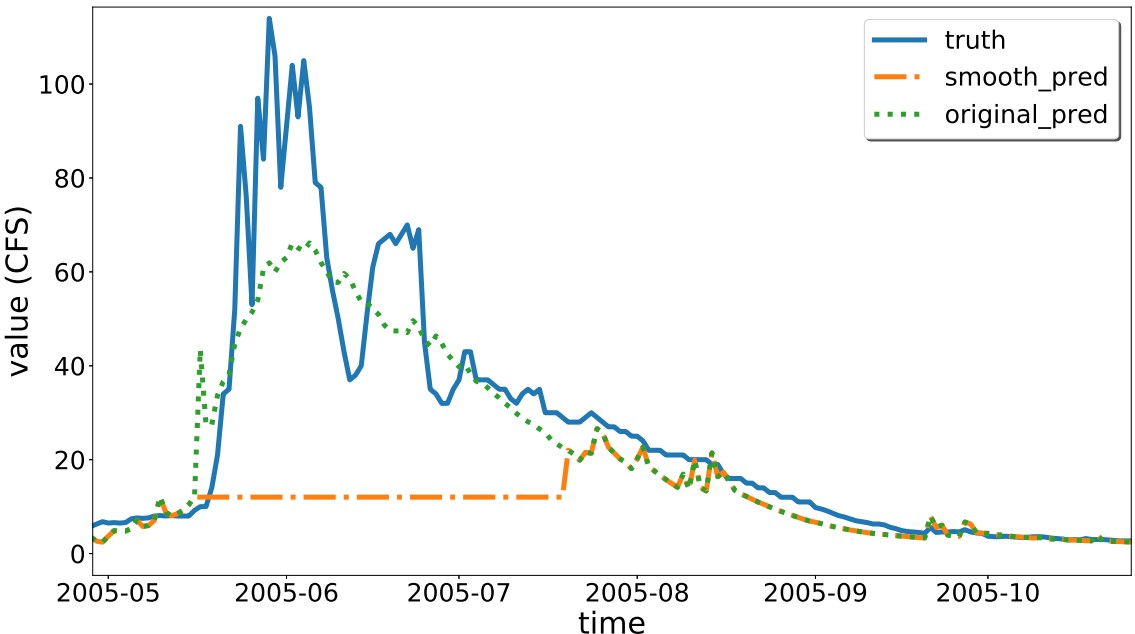

**Figure 9.** Use 10 as Threshold: There is a plateau around 2005 June generated. CFS is short for Cubic Feet per Second

## 3 Results and Analysis

### 3.1 Experiment Design

Each dataset is separated into training dataset (50%) and testing dataset (50%). We use the quantitative statistics to perform the statistical evaluation of modeling accuracy in the testing step: RMSE, PBIAS, NSE, and CD. The statistical parameters are defined by the following equations:

$$RMSE = \sqrt{\frac{1}{N}\sum_{i=1}^{N}(P_i - A_i)^2} \tag{8}$$

$$PBIAS = \frac{\sum_{i=1}^{N}(A_i - P_i)100}{\sum_{i=1}^{N}A_i} \tag{9}$$



$$NSE = 1 - \frac{\sum\limits_{i=1}^{N}(A_i - P_i)^2}{\sum\limits_{i=1}^{N}(A_i - \bar{A})^2} \tag{10}$$

$$CD = \left\{ \frac{\sum\limits_{i=1}^{N}(A_i - P_i)(P_i - \bar{P})}{\left(\sum\limits_{i=1}^{N}(A_i - \bar{A})^2\right)^{\frac{1}{2}}\left(\sum\limits_{i=1}^{N}(P_i - \bar{P})^2\right)^{\frac{1}{2}}} \right\}^2 \tag{11}$$

where $P_i$ and $A_i$ represent the simulated and observed values respectively; $\bar{A}$ is the mean of the observed values and $\bar{P}$ is the mean of simulated values for the entire evaluation period.

RMSE measures how close the observed data are to the predicted values while retaining the original units of the model's output and observed data. Lower values of RMSE indicate a better fit of the model. RMSE is one of the important standards that defines how accurately the model predicts the response and it is commonly used in many fields.

PBIAS is a measure to evaluate the model simulations. It determines whether the predictions are underestimated or overestimated, compared to the actual observations. If the PBIAS values are positive, the model overestimates the results; otherwise, the model underestimates the results by the given percentage. Therefore, values closer to zero are preferred for PBIAS.

The Nash-Sutcliffe Efficiency (NSE) is a normalized statistic assessing the model's ability to make predictions that fit $1:1$ line with the observed values. The values for NSE range between $-\infty$ and 1. For acceptable levels of performance, the values of NSE should lie close to one, and the higher NSE indicates the better results.

CD stands for coefficient of determination, calculated as the square of the correlation between the observed values and the simulated values. The values for CD ranges between 0.0 and 1.0 and correspond to the amount of variation in the simulated values (around its mean) that is explained by the observed data. Values closer to one indicate a tighter fit of the regression line with the simulated data. Similar to NSE, the higher CD values indicates the better results.

In the following case studies, we also provide Prediction Interval (PI) which offers the possible prediction range. The PI is calculated using Eq. 12, where $\bar{X}$ is the sample mean, $n$ is the number of samples, $T_a$ is student's t-distribution percentile with $n-1$ degrees of freedom. PI is described with upper bound and lower bound.

$$PI = \bar{X}_n \pm T_a s_n \sqrt{1 + (1/n)} \tag{12}$$

## 3.2 Case Study 1

### 3.2.1 The PRMS Hydrologic Model

The Precipitation-Runoff Modeling System (PRMS) was developed by U.S. Geological Survey in the 1980s, which is a physically based parameter-distributed hydrologic modeling system Leavesley et al. (1983); Markstrom et al. (2005, 2015). The





PRMS model used in this study was developed by Chen et al. Chen et al. (2015) in the study area of Lehman Creek watershed, eastern Nevada. The watershed is located in the Great Basin National Park, occupying an area of 5,839 acres of the southern Snake Valley Prudic et al. (2015); Volk (2014). More than 78% of land cover were evergreen forest, deciduous forest, and mix forest, 2% of shrubs, 2% were perennial snow and ice, and 17% were barren land Chen et al. (2016, 2015). The streamflow is mainly composed by snowmelt, which sourced from the high elevated area in the west, flowing over the large mountain quartzite and recharging the groundwater system through alluvial deposits and karst-limestone in the east Chen et al. (2017). These high hydro-geography variations made it appropriate to use PRMS model to describe the spatial heterogeneity of hydrologic processes. Fig. 10 displays the study area.

On a grid-based simulation, the Lehman Creek watershed was delineated by 96 columns and 49 rows using 100 x 100-meter cell/grid. A total number of 4074 grids were formed, and based on which, the combinative effects of canopy interception, evapotranspiration, infiltration, overland runoff, and subsurface flow were simulated. The parameter estimation is one of the most critical and challenging parts of the PRMS model development. They were estimated for model algorithms and determining the model performance, using land cover land use, soil information or through literatures for each hydrologic component on each of 4704 units Chen et al. (2015). Among all the parameters required for model runs, some parameters are specifically sensitive and have great influences on the model simulation results. Such as parameters that determine the temporal and/or spatial distribution of precipitation, requires specification on every one of 12 months and/or every one of 4704 cells (e.g., $tmax\_allsnow$, monthly maximum air temperature when precipitation is assumed to be snow; $snow\_adj/rain\_adj$, monthly factor to adjust measured precipitation on each HRU to account for differences in elevation, and so forth; $tmin\_lapse$, monthly values representing the change in minimum air temperature per 1,000 $elev\_units$ of elevation change).

One station meteorologic data were used as the driving forces to the developed model in the study area of Lehman Creek watershed. Daily precipitation, maximum temperature, and minimum temperature from October 1, 2003 to September 30, 2012 were collected from the meterologic station (#263340, Great Basin NP). Daily streamflows at the Lehman CK Nr Baker gauging station (#10243260) were collected for model calibration and validation Chen et al. (2015).

### 3.2.2 Results

The goal is to improve the PRMS model streamflow predictions. First, the training dataset is transformed by using log-sinh transformation, which is introduced in Wang et al. (2012). Eq. 13 is the transformation equation and Eq. 14 is the back transformation equation.

$$\hat{y} = \frac{log(sinh[a+by])}{b} \tag{13}$$

$$y = \frac{sinh^{-1}(10^{\hat{y}b}) - a}{b} \tag{14}$$

where $a$ and $b$ are transformation parameters. By using log-sinh transformation, the original randomly distributed errors are normalized for the convenience of correlation characterization.





**Figure 10.** PRMS Hydrologic Model study Area.




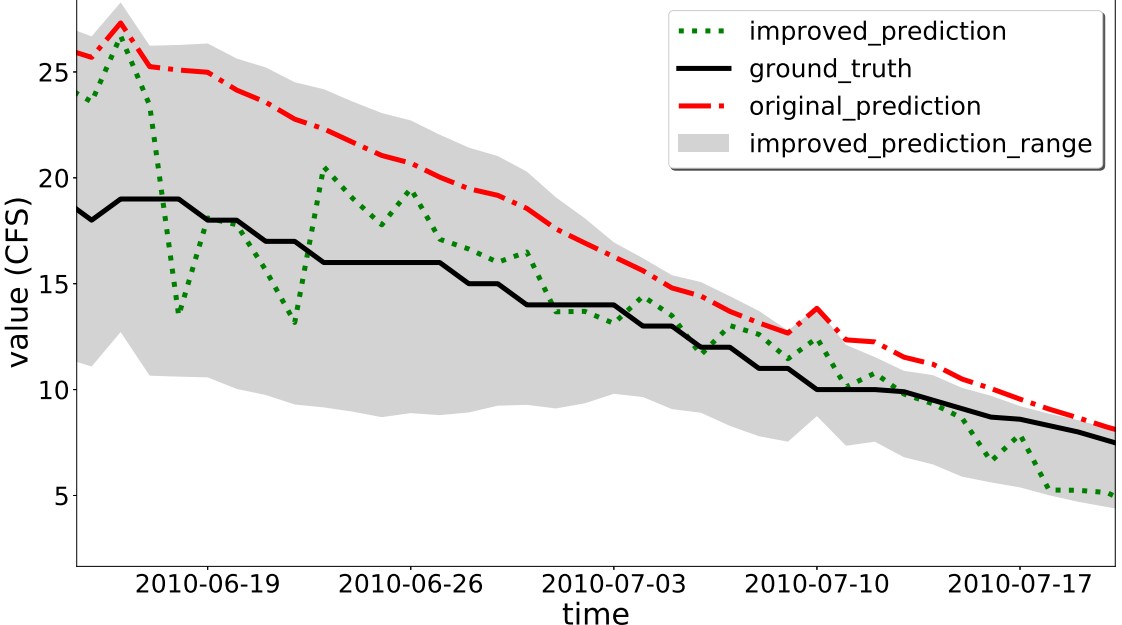

**Figure 11.** Case Study 1 Final PRMS Model Streamflow Prediction Improvements. The improved predictions are closer to the ground truth than the original predictions.

During the training process as evaluated by using cross validation, we found the best scale factor $\alpha$ is 0.5, the best transformation parameter $a$ is 0.0305 and $b$ is 0.0605, where $\alpha$ is used in Eq. 2; $a$ and $b$ are used in Eq. 13. Gradient Boosted Trees Hastie et al. (2009) is used in the "Machine Learning Model" component and the initial window size is one-year.

Note that we find that the improved PRMS model predictions do not well follow the observations during the water recession
5   period after the peak flow. This is caused by unstable historical data. By using the smooth method introduced in Section 2.3, the RMSE is further improved to be 2.032 with $T = 10$. The comparisons between parts of the data are shown in Fig. 11. It is clear that the improved predictions are closer to the ground truths than the original PRMS predictions. All the statistical measurement results summarized are shown in Table 1. As results show, the improved predictions have lower RMSE indicating they are closer to the observed data. The PBIAS value is larger than the original PRMS model suggesting an over-estimation
10   compared with the observations. The NSE value is closer to one, which means the improved model has a more acceptable level of performance. The CD value is closer to one means the improved model fits more to the observations. As suggested by the comparison results of model performance evaluation indicators, the proposed framework can improve the original PRMS model's results.



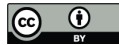

**Table 1.** Calibrated PRMS Model Results Comparisons.

| | Indicators | | | |
|---|---|---|---|---|
| **Model** | **RMSE** | **PBIAS** | **CD** | **NSE** |
| Original PRMS | 4.585 | _**7.205**_ | 0.769 | 0.768 |
| Improved PRMS | _**2.032**_ | 10.808 | _**0.936**_ | _**0.926**_ |

**Table 2.** Uncalibrated PRMS Model Results Comparisons.

| | Indicators | | | |
|---|---|---|---|---|
| **Model** | **RMSE** | **PBIAS** | **CD** | **NSE** |
| Original PRMS | 8.439 | -82.658 | 0.001 | -0.292 |
| Improved PRMS | _**3.092**_ | _**3.054**_ | _**0.837**_ | _**0.826**_ |

As suggested by statistical measurement comparisons in Table 2, our proposed framework can also improve uncalibrated PRMS model predictions. With the same PRMS model and input data, the RMSE is improved from 8.439 to 3.092 by using 1.0, 0.0905, 0.0805, and 10 for $\alpha$, a, b, and the smooth threshold respectively. The RMSE is very close to the improved calibrated model RMSE (2.032), which indicated the proposed framework can possibly be an effective replacement the traditional
complex time-consuming calibration procedure, providing a competitive level of model performance.

### 3.3 Case Study 2

#### 3.3.1 Hydrologic Modeling System

The Hydrologic Modeling System (HEC-HMS), released by U.S. Army Corps of Engineers in 1998, is designed to simulate the hydrologic processes of dendritic watershed system Bennett (1998); Scharffenberg and Fleming (2006). Different from
the PRMS model that focuses on the hydrologic components based on user-defined unit, the HEC-HMS uses a dendritic-based precipitation-runoff model with integrations in water resources utilization, operation, and management Scharffenberg and Fleming (2006). The case study of HEC-HMS was the Little River Watershed, which is an example application model in HEC-HMS program for the demonstration of the continuous simulation with the soil moisture accounting method Bennett and Peters (2004). As introduced by Bennett and Peters Bennett and Peters (2004), the Little River Watershed is a 12,333-acre
(19.27 $m^2$) basin near Tifton, Georgia. More than 50% of the land is covered by forest with remaining land used for agricultural purposes USDA (1997). The annual precipitation is 48 inches Center. (1998).

One single-station data of precipitation observation were used, which was from the Agricultural Research Service (ARS) rain gauge (#000038) Georgia (2007). The precipitation records were on a 15-min basis for the same model running period of January 1 1970-Jun 30 1970. The streamflow observations were from ARS gauge #74006 Georgia (2007) on an hourly basis,
which were used for the calibration and validation of this hydrologic model performance.





**Table 3.** Calibrated HEC-HMS Model Results Comparisons.

|  | Indicators | | | |
|---|---|---|---|---|
| **Model** | **RMSE** | **PBIAS** | **CD** | **NSE** |
| HEC-HMS | 44.983 | _**4.657**_ | 0.842 | 0.808 |
| Improved HEC-HMS | _**39.844**_ | 8.590 | _**0.884**_ | _**0.850**_ |

### 3.3.2  Results

In case study 2, the goal is to improve the HEC-HMS streamflow predictions. We use Boxcox transformation Wang et al. (1964) to transform the dataset and choose decision tree in the "Machine Learning Model" component to improve the hydrologic model accuracy. Boxcox transformation is simple but efficient method and able to reduce dataset variances. A decision tree consumes
much less time than most machine learning methods (such as gradient boosted trees) with the same inputs in the training phase. Eq. 15 is the Boxcox transformation equation and Eq. 16 is the back Boxcox equation function.

$$\hat{y} = \frac{y^{\lambda} - 1}{\lambda} \tag{15}$$

$$y = \sqrt[\lambda]{\hat{y}\lambda - 1} \tag{16}$$

where $\lambda$ is the transformation parameter. During the training process as indicated by using cross validation, the best $\alpha$ is 0.3
and the best $\lambda$ is 9.0 for this case study. The window size of one-week is selected. By using our proposed method, the RMSE is 39.844 compared to 44.9833 resulting from the original HEC-HMS PRMS model. Fig. 12 shows the prediction comparisons of parts of the data between the lower bound, upper bound, improved prediction, ground truth, and original prediction. Clearly, the improved prediction is more accurate than the original hydrologic model predictions.

As summarized in Table 3, RMSE of the improved model is 39.844 and it is lower than the original HEC-HMS RMSE
(44.983), which means the outputs are closer to the observed data. PBIAS (4.657) of the original model is closer to zero than the improved HEC-HMS PBIAS (8.590), which means the improved method over-estimates the observations. The NSE and CD values (0.850 and 0.884) of the improved HEC-HMS are closer to one than the original HEC-HMS values (0.808 and 0.842), which means the improved model has a more acceptable level of performance and fits more to the observations. The smooth method, which is introduced in Section 2.3, cannot improve the results. This because there are not many spikes along
the uphills and downhills.

As suggested by the statistical measurement comparisons in Table 4, the proposed method can also improve the uncalibrated HEC-HMS model. By inputting the same data, the RMSE is reduced from 134.610 to 89.882 by using 0.8 and 11 for $\alpha$ and $\lambda$ respectively. The time window is one-week.





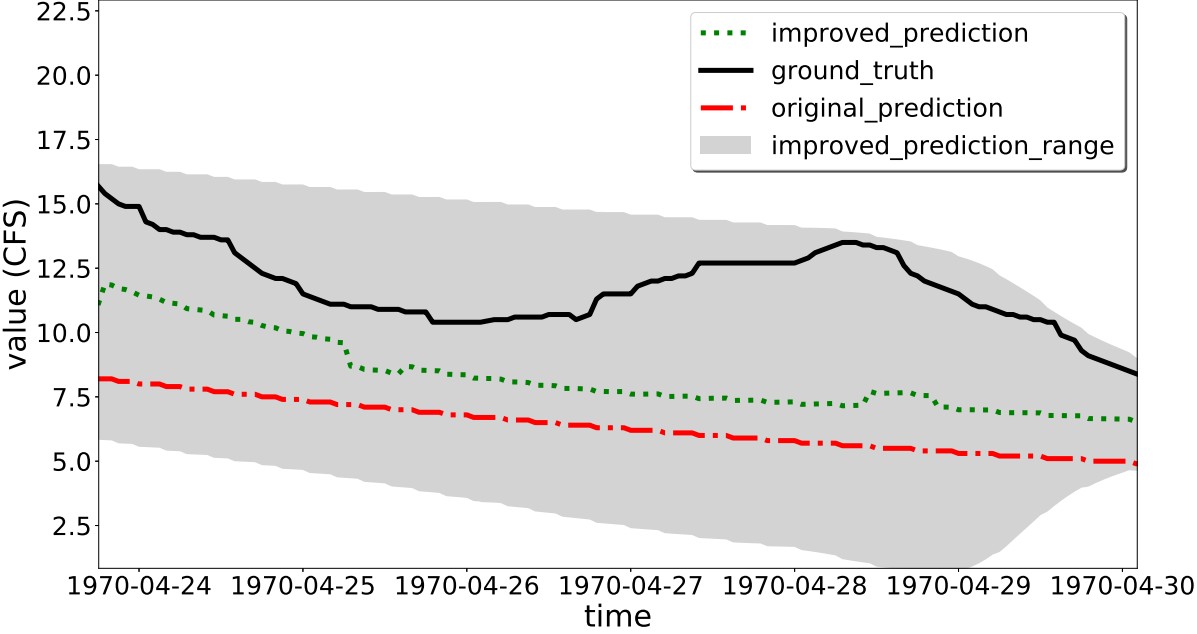

**Figure 12.** Case Study 2 HEC-HMS PRMS Model Improvements. The improved predictions are closer to the ground truth than the original predictions.

**Table 4.** Uncalibrated HEC-HMS Model Results Comparisons.

| | Indicators | | | |
|---|---|---|---|---|
| **Model** | **RMSE** | **PBIAS** | **CD** | **NSE** |
| HEC-HMS | 134.610 | 45.943 | 0.768 | -0.716 |
| Improved HEC-HMS | *89.882* | *29.876* | *0.823* | *0.235* |

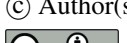



## 4  Discussion

The current study used two typical hydrologic models, PRMS and HEC-HMS, and demonstrated the performance of the proposed post-processor framework. To have a comprehensive evaluation, these two models are selected as representations from hydrologic models categories that differentiate in terms of simulation scopes, structures, and applications. As a representation

of physically-based parameter-distributed hydrologic models, PRMS is widely used for research purposes, which requests large sets of parameters to simulate the physical processes; comparatively, as a representation of empirical-based lumped-parameter hydrologic models, HEC-HMS is widely used in industrial engineering purposes, which conceptualized physical bases towards result-oriented simulation.

While implementing the pre-developed hydrologic simulation, the calibrated hydrologic models were "restored" to the orig-

inal uncalibrated status for a comparison purpose. During the "restoration", the calibrated parameters were adjusted to default values either from program manuals or authors' personal suggestions. This may lead to a varying "restoration" status of uncalibrated model performance depend on parameters suggested. However, in this study, the main goal for the development of uncalibrated hydrologic models is to compare model simulation/post-processing performance in a qualitative sense. Thus, the detail of uncalibrated model development is not the main focus in the study.

There is one thing should be aware of in the PRMS simulation of the Lehman Creek watershed. According to Prudic et al. (2015), during 2011 summer, the peak flow observation was under-recorded due to the large overland flow bypassing the gauge station. The actual peak flow rate should be as great as the peak flow rate in 2005, since the precipitation in these two years are comparable. However, the current calibrated PRMS model was not able to capture the actual high peak flow but the observed peak flow. Nevertheless, this results in a better fitness with observations instead of over estimation and making the fitness

evaluation in PRMS model and post-processor more comparable.

## 5  Conclusion

In this paper, a post-processor framework is proposed to improve the accuracy of hydrologic models with a window size selection method embedded to solve the non-stationary concern in hydrologic data. The proposed post-processor framework leverages machine learning approaches to characterize the role that the model inputs play in the model prediction errors so

as to improve hydrologic model prediction results. The proposed window size selection method enhances the performance of the proposed framework when dealing with non-stationary data. The results of two different hydrologic models show that the accuracy of calibrated hydrologic models can be further improved; without efforts of the calibration, the results of uncalibrated hydrologic models using the proposed framework can be as accurate as the calibrated ones by leveraging the proposed framework, which means that our proposed methods are possibly able to ease the traditional complex and time-consuming model

calibration step.

Two case studies are introduced in this paper and we will exam the framework with other models and study fields. Also, it is interesting to study the peak values and better prediction algorithm for the peak values in the future.

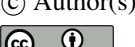



## 6   Code and Data Availability

The code and data introduced in this paper are available in Wu (2018). The prototype program is mainly written in Python and more instructions can be found in the repository readme file. The data files are stored in the "data" folder and generated by hydrologic models introduced in Section 3.2.1 and Section 3.3.1.

*Acknowledgements.*   This material is based upon work supported by the National Science Foundation under grant numbers IIA-1329469 and IIA-1301726 and also University of Nevada, Reno Graduate Student Association Research Grand Program. Any opinions, findings, and conclusions or recommendations expressed in this material are those of the authors and do not necessarily reflect the views of the National Science Foundation.

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
