# Peer review of "MELPF Version 1: Modeling Error Learning based Post-Processor Framework for Hydrologic Models Accuracy Improvement"

_Geoscientific Model Development, 2018_

## Short Comment (SC1) · 15 Aug 2018

Dear authors,

in my role as Executive editor of GMD, I would like to bring to your attention our Editorial version 1.1:

http://www.geosci-model-dev.net/8/3487/2015/gmd-8-3487-2015.html

This highlights some requirements of papers published in GMD, which is also available on the GMD website in the 'Manuscript Types' section:

http://www.geoscientific-model-development.net/submission/manuscript_types.html

In particular, please note that for your paper, the following requirements have not been

met in the Discussions paper:

- "The main paper must give the model name and version number (or other unique identifier) in the title."

- "If the model development relates to a single model then the model name and the version number must be included in the title of the paper. If the main intention of an article is to make a general (i.e. model independent) statement about the usefulness of a new development, but the usefulness is shown with the help of one specific model, the model name and version number must be stated in the title. The title could have a form such as, "Title outlining amazing generic advance: a case study with Model XXX (version Y)"."

In order to simplify reference to your Post-Processing Framework, please add a name (and/or its acronym) and a version number in the title of your article in your revised submission to GMD.

Additionally, please provide details how to access the two hydrological models used in your study.

Yours,

Astrid Kerkweg

———————————————————

---

## Short Comment (SC2) · 10 Sep 2018

Dear Rui Wu,

thanks for adding the required information to your manuscript!

However, it would be good to include the references where to download the hydrological models in the "Code and Data availability" section. Here you only cite Wu (2018), which is somehow confusing (at least also refer to Sect. 4, better provide the web-links).

Regards,
Astrid Kerkweg
* * *
[Figure]

2018.

---

## Referee Comment (RC1) · William Paul Miller (Referee) · 17 Apr 2019

I find the work presented here interesting and informative. The work presented here is thought provoking and offers another way to approach hydrologic modeling. I have a couple comments:

On page 7, the authors note that the uncalibrated models with default values are compared with the calibrated cases from traditional calibration and post-processor methods. To what extent do the default values impact the results of this study? Are the default values relatively close to the traditionally calibrated values, or are they significantly different? Do the default values accurately, or inaccurately, physically represent the system being modeled. I think it's important for the authors to discuss how the

default values in the models studied here impact the results presented.

Are the authors able to attribute errors identified through their Modeling Error Learning algorithm to any model biases that are physically based? For example, is there a particular streamflow behavior or weather pattern that is not accurately captured in the HEC-HMS or PRMS models due to a limitation into how a physical process is modeled or represented in those models? If there is no attributable physical reason for the errors identified through the learning algorithm, is it really appropriate to be making those changes; that is, are we getting the right (or more accurate) answer for the wrong reason? The modeling Error Learning algorithm may be just identifying a limitation of the model.

---

## Author Comment (AC3) · 13 May 2019

Dear Dr. William Paul Miller,

Thank you for your comments. We have carefully reviewed the comments and have revised the manuscript accordingly. Our responses are given in a point-by-point manner below. Changes to the manuscript are shown in bold.

âŮŔOn page 7, the authors note that the uncalibrated models with default values are compared with the calibrated cases from traditional calibration and post-processor methods. To what extent do the default values impact the results of this study? Are the default values relatively close to the traditionally calibrated values, or are they significantly different? -Response: Thanks for the comment. A physical hydrologic model

usually cannot generate good results with default values. In the paper, we have two examples that show default values produce inaccurate results by comparing Table 1 Calibrated Orignal PRMS vs. Table 2 Uncalibrated Orignal PRMS and Table 3 Calibrated Original HEC-HMS vs. Table 4 Uncalibrated Original HEC-HMS are good examples. To further explain our ideas, we added the following sentences in the Discussion section: "A physical hydrologic model usually cannot generate good results with default values. In the paper, we have two examples that show default values produce inaccurate results. With the same model and study area, Table 1 Calibrated Orignal PRMS results are much more accurate than Table 2 Uncalibrated Orignal PRMS based on performance evaluation indices. Similarly, Table 3 Calibrated Original HEC-HMS results are much better than Table 4 Uncalibrated Original HEC-HMS. We do have conducted experiments to test different values of default settings on the modeling performance and our proposed can improve the performance."

âŮŔDo the default values accurately, or inaccurately, physically represent the system being modeled. I think it's important for the authors to discuss how the default values in the models studied here impact the results presented. -Response: We appreciate the comment. This is a great point. To clarify our ideas, we added the following sentences in the Discussion section: "There may be various types of default parameters used in a physical hydrologic model for development efficiency. Parameters can be classified as sensitive and insensitive, or model execution-related and process algorithm related. Apart from the model execution-related parameters and other insensitive parameters, those process algorithm related sensitive parameters are typically critical to model development, which greatly affect the model performance. Default values can be well contained in physical laws and corresponding computation algorithms, but not necessarily capture the regional hydrologic characteristics at a study site. Capturing such site-specific features is the process of calibration. As such, the differences between uncalibrated - default set- models and calibrated models are determined by the significance of sensitive parameters in affecting the modeling performance."

âŮŔAre the authors able to attribute errors identified through their Modeling Error Learning algorithm to any model biases that are physically based? For example, is there a particular streamflow behavior or weather pattern that is not accurately captured in the HEC-HMS or PRMS models due to a limitation into how a physical process is modeled or represented in those models? -Response: Thank you for the comment. The proposed method is not physically based. It is tested with physical hydrologic models. We have done experiments and found that if the errors of hydrologic models are correlated with model inputs, our proposed method will work. We have mentioned this point in the abstract. To further clarify the idea, the following paragraph is added to Section 4 Discussion: "The proposed method is not designed specifically for physically based models. We tested the proposed methods with physical hydrologic models and would like to exam it with other types of models in the future. Based on our opinion, the proposed method works because it can find hydrologic model limitations based on the patterns of model errors."

âŮŔIf there is no attributable physical reason for the errors identified through the learning algorithm, is it really appropriate to be making those changes; that is, are we getting the right (or more accurate) answer for the wrong reason? The modeling Error Learning algorithm may be just identifying a limitation of the model. -Response: Thank you for pointing this out. Yes, our proposed method is based on the limitations of a hydrologic model. To clarify our idea, the following paragraph is added to Section 4 Discussion: "Our proposed method identifies the limitations of a physical hydrologic model based on errors and its correlation with model inputs. If there is such a connection between model errors and inputs, it means the hydrologic model does not characterize the relation between inputs and outputs well enough. To fix the issue, we leverage machine learning techniques and propose a novel method to find out data patterns in this paper."

Sincerely, Rui Wu, Lei Yang, Chao Chen, Sajjad Ahmad, Sergiu M. Dascalu, and Frederick C. Harris, Jr.

Please also note the supplement to this comment:
https://www.geosci-model-dev-discuss.net/gmd-2018-136/gmd-2018-136-AC3-supplement.pdf

———————————————————

[Figure]

**Supplement:**

[revised manuscript text omitted]

---

## Author Comment (AC4) · 14 May 2019

Dear Dr. William Paul Miller,

Thank you for your comments. We have carefully reviewed the comments and have revised the manuscript accordingly. Our responses are given in a point-by-point manner below. Changes to the manuscript are shown in bold in this paper.

Review: On page 7, the authors note that the uncalibrated models with default values are compared with the calibrated cases from traditional calibration and post-processor methods. To what extent do the default values impact the results of this study? Are the default values relatively close to the traditionally calibrated values, or are they significantly different? -Response: Thanks for the comment. A physical hydrologic model

usually cannot generate good results with default values. In the paper, we have two examples that show default values produce inaccurate results by comparing Table 1 Calibrated Original PRMS vs. Table 2 Uncalibrated Original PRMS and Table 3 Calibrated Original HEC-HMS vs. Table 4 Uncalibrated Original HEC-HMS. To further explain our ideas, we added the following sentences in the Discussion section: "A physical hydrologic model usually cannot generate good results with default values and requires calibration (Chen30et al., 2015b; Hay et al., 2006; Hay and Umemoto, 2007b). In the paper, we have two examples that show default values produce inaccurate results. With the same model and study area, Table 1 Calibrated Original PRMS results are much more accurate than Table 2 Uncalibrated Original PRMS based on performance evaluation indices. Similarly, Table 3 Calibrated Original HEC-HMS results are much better than Table 4 Uncalibrated Original HEC-HMS. Numerical experiments have corroborated the superior performance of the proposed method, compared with traditional methods with different default values."

Review: Do the default values accurately, or inaccurately, physically represent the system being modeled. I think it's important for the authors to discuss how the default values in the models studied here impact the results presented. -Response: We appreciate the comment. This is a great point. To clarify our ideas, we added the following sentences in the Discussion section: "There may be various types of default parameters used in a physical hydrologic model for development efficiency. Parameters can be classified as sensitive and insensitive, or model execution related and process algorithm related. Apart from the model execution related parameters and other insensitive parameters, the process algorithm related sensitive parameters are typically critical to model development, which greatly affect the model's performance. Default values can well follow physical laws and be contained in the corresponding computation algorithms, but not necessarily capture the regional hydrologic characteristics at a study site. Capturing such site-specific features is the process of calibration. As such, the differences between uncalibrated - default set- models and calibrated models are determined by the significance of sensitive parameters in affecting the modeling performance."

Review: Are the authors able to attribute errors identified through their Modeling Error Learning algorithm to any model biases that are physically based? For example, is there a particular streamflow behavior or weather pattern that is not accurately captured in the HEC-HMS or PRMS models due to a limitation into how a physical process is modeled or represented in those models? -Response: Thank you for the comment. The proposed method is data-driven and not only designed for physically based models. It is tested with physical hydrologic models. We have done experiments and found that if the errors of hydrologic models are correlated with model inputs, our proposed method will work. We have mentioned this point in the abstract. To further clarify the idea, the following paragraph is added to Section 4 Discussion: "The proposed method is not designed specifically for physically based models. We tested the proposed methods with physical hydrologic models and would like to exam it with other types of models in the future. In our opinion, the proposed method works because it can find hydrologic model limitations, such as improve modeling peak values, based on the patterns of model errors."

Review: If there is no attributable physical reason for the errors identified through the learning algorithm, is it really appropriate to be making those changes; that is, are we getting the right (or more accurate) answer for the wrong reason? The modeling Error Learning algorithm may be just identifying a limitation of the model. -Response: Thank you for pointing this out. Yes, our proposed method is based on the limitations of a hydrologic model. The limitations of HEC-HMS and PRMS models are that these two models cannot model peak values. Figure 2 is an example to show that a hydrologic model's error is much higher if there is a peak. To clarify our idea, the following paragraph is added to Section 4 Discussion: "In this paper, model limitations mean peak values. For example, if a hydrologic parameter changes massively within a short period, i.e., peak values, a physical hydrologic model may not be able to characterize the trend. Figure 2 is an example that shows a physical hydrologic model has a higher

error rate when there is a peak. Our proposed method identifies the limitations of a physical hydrologic model based on errors and their correlation with model inputs. If there is such a connection between model errors and inputs, it means the hydrologic model does not characterize the relation between inputs and outputs well enough. To fix the issue, we leverage machine learning techniques and propose a novel method to find out data patterns in this paper." Sincerely, Rui Wu, Lei Yang, Chao Chen, Sajjad Ahmad, Sergiu M. Dascalu, and Frederick C. Harris, Jr.

Please also note the supplement to this comment:
https://www.geosci-model-dev-discuss.net/gmd-2018-136/gmd-2018-136-AC4-supplement.pdf

**Supplement:**

[revised manuscript text omitted]

---

## Referee Comment (RC2) · Bethanna Jackson (Referee) · 3 Jun 2019

As handling topical editor for this paper, I have struggled to get reviewers for it much more than normal (first to accept,and then if accepting, to submit). My take on the paper when accepting to handle it was that it appeared to be high quality and interesting/thought provoking, which is backed up by the one constructive, informed and positive review we did secure (thank you, Paul Miller; authors, please do acknowledge his constructive comments in your final manuscript). I did anticipate it might be difficult to secure reviewers as it is bringing an unsolicited perspective informed by disciplines beyond those that traditionally contribute to the already contested topic of "how best to handle errors in hydrological modelling" debate. I hadn't anticipated it would be quite as difficult as it turned out to be to get reviews. I think it's taking people outside their

comfort zone, and that's a good thing.

Given my initial take on its quality, backed up by a more recent read, the reviewer comments and response from the authors, I am happy to recommend publication. I don't want a paper proposing a new perspective and approach with promise, whether or not that promise works out, to be held up any further by our continuing struggle in the academic community to find time for multiple peer review of an interesting but complex and somewhat out of left field proposal.

Before it proceeds to publication, I would ask the authors to reconsider their response to Paul's last comment and addressing it in the publication, as I'm not sure you fully got the gist of it (your responses to all the others were great, however).

Paul's final point was: "If there is no attributable physical reason for the errors identified through the learning algorithm, is it really appropriate to be making those changes; that is, are we getting the right (or more accurate) answer for the wrong reason? The modeling Error Learning algorithm may be just identifying a limitation of the model."

Your response to this just targets you are addressing limitations of the model in your approach. My take on this comment is that he was acknowledging the value machine learning algorithms could bring to hydrology but also stressing the importance, for it to be really valuable, of not just relying on mathematical/computer science data mining but also process understanding. This has been a hard learned lesson in hydrology.

The finding that errors of hydrological models are most strongly correlated with model inputs that you note in the abstract, is correct but already very well understood and acknowledged. We have had that understanding at least the last two decades, and recognize this often leads to our models sometimes compensating with structural and/or parameterisation errors that at least partially compensate for these biases. As data quality changes, these biases changes and our models aren't necessarily updated with structural changes that reflect that, which effects predictive power, and the models fitting the data are also used for hypothesis testing to understand dominant process

better, so model structural compensations to address input bias hold our scientific understanding back. A pressing issue in hydrology.

I'd like to see a couple of paragraphs to recognise that in the paper. I do compliment you on the worth of the paper, but that issue of disentangling biassed input data and biassed parameters (all inputs from a mathematical sense, but distinct in that the parameters are related to structure) and issues with model structure are not solved by the approach, and that an extension to recognise and potentially help disentangle these to some extent would bring added value.

---

## Author Comment (AC5) · 18 Jun 2019

Dear Dr. Bethanna Jackson,

Thank you for your comments. We have carefully reviewed the comments and have revised the manuscript accordingly. Our responses are given in a point-by-point manner below. Changes to the manuscript are shown in bold.

âŮŔ We acknowledged Dr. Paul Miller's constructive comments in the paper with the following text: "We really appreciate valuable comments from all the reviewers, especially Dr. Paul Miller for his constructive comments."

âŮŔ Thank you for pointing out Dr. Paul Miller's last comment. To clarify our ideas from both computer science and hydrologic science, we added the following paragraphs in

the Discussion Section: " As model driving forces, the data input is heavily relied upon in physical-based hydrologic models. On physical bases, the meteorologic input is modeled with water flow storage and path within the earth system. The streamflow, as demonstrated in this research, is one of the examples. During this process, all numerical models simplify physical processes to some degree, either spatial-wise, such as hydrologic response unit, or temporal-wise, such as summer leaf index. Such conceptualization and simplification compose a static numerical modeling environment that cannot capture all environmental stressors, such as in the meteorological inputs. This is long-time stressing issues in hydrologic science.

To capture the environmental stressors, such as meteorological changing trend, land cover variation, vegetation growth, we can use different hydrologic models or add additional physical-based algorithms to capture the specific processes and correct bias from missing representations. However, with a mix of stressor, it is hard to distinguish the causes of biases and remove/mitigate these biases, from data input, parameters or model structures. Machine learning techniques fill this gap.

Instead of switching to another model better capturing data input, according to our experiment results, the proposed machine learning techniques help update a hydrologic model to characterize input data bias as a plug-in in our proposed framework. It can sense data trend and compensate hydrologic model predictions with the window selection method. The effect is similar to have multiple hydrologic models for different input data biases.

Machine learning in this application attempts to use relevant input data to reproduce hydrologic behavior, i.e., flow hydrograph as close to observed as possible. The overall difference in observed and modeled hydrograph is categorized as an error. In hydrologic literature, it has been recognized that this difference can be due to uncertainty in input and output data, bias in model parameterization, and issues with model structure. With the current machine learning approaches, it is not possible to disentangle and attribute total error to multiple sources such as input data, model parameters, and model

structure. Moreover, machine learning approaches cannot provide physical reasoning for this error. This is a recognized issue in hydrology and an active area of research. Since no prior model structure is provided to machine learning approach - it learns model structure and parameters from input data and observed output- it can be stated that contribution of model structure and parameters towards total error is relatively small compared to bias or uncertainty in model input. The separation of data into training and testing samples provides a safeguard against overfitting the model. However, issue of disentangling error and attributing it to multiple sources remains unresolved in this work. Future research should focus on this issue."

Sincerely, Rui Wu, Lei Yang, Chao Chen, Sajjad Ahmad, Sergiu M. Dascalu, and Frederick C. Harris, Jr.